# *Fructus lycii*: A Natural Dietary Supplement for Amelioration of Retinal Diseases

**DOI:** 10.3390/nu13010246

**Published:** 2021-01-16

**Authors:** Kumari Neelam, Sonali Dey, Ralene Sim, Jason Lee, Kah-Guan Au Eong

**Affiliations:** 1Department of Ophthalmology and Visual Sciences, Khoo Teck Puat Hospital, Singapore 768828, Singapore; lee.jason.ks@ktph.com.sg (J.L.); aekg@eyecataractretina.com (K.-G.A.E.); 2Singapore Eye Research Institute, Singapore 169856, Singapore; 3Yong Loo Lin School of Medicine, National University of Singapore, Singapore 119228, Singapore; e0105134@u.nus.edu (S.D.); ralene_sim1995@hotmail.com (R.S.); 4International Eye Cataract Retina Center, Farrer Park Medical Center, Singapore 217562, Singapore

**Keywords:** *Fructus lycii*, carotenoids, antioxidants, age-related macular degeneration, diabetic retinopathy, retinitis pigmentosa

## Abstract

*Fructus lycii* (*F. lycii*) is an exotic “berry-type” fruit of the plant *Lycium barbarum* that is characterized by a complex mixture of bioactive compounds distinguished by their high antioxidant potential. *F. lycii* is used in traditional Chinese home cooking and in the Chinese Pharmacopeia as an aid to vision and longevity as well as a remedy for diabetes to balance “*yin*” and “*yang*” in the body for about two centuries. Although a myriad of bioactive compounds have been isolated from *F. lycii*, polysaccharides, carotenoids, flavonoids, and phenolics represent the key functional components of *F. lycii*. *F. lycii* has been shown to exhibit a wide range of biological activities in experimental settings including antioxidant, anti-inflammatory, antiapoptotic, and neuroprotective effects. Despite its medicinal role dating back to the eighteenth century in the Far East and robust evidence of beneficial effects on ocular health and retinal diseases originating mainly from studies in animal models, the role of *F. lycii* in the clinical management of retinal diseases is yet to be established. This article comprehensively reviews the literature germane to *F. lycii* and retinal diseases with particular emphasis on age-related macular degeneration, diabetic retinopathy, and retinitis pigmentosa, which are commonly seen in clinical practice.

## 1. Introduction

*Fructus lycii* (*F. lycii*) is a “berry-type” fruit of the plant *Lycium barbarum,* which is a deciduous shrub belonging to the family Solanaceae. Although *Lycium* fruit was described in the *Shennong Ben Cao Jing* (ca 100AD), the oldest surviving Chinese *materia medica*, Carl Linnaeus, a Swedish botanist, provided the genus name ”*Lycium* (L)“ in the year 1753 and gave the species name ”*barbarum*” [1,2]. Commonly known as goji berry, *F. lycii* is also known as wolfberry in English, *kei tze* in Cantonese, and g*ou qi zi* in Mandarin (z*i* meaning seed or specifically berry). Fresh *F. lycii* is bright orange-red in color with an ellipsoid shape measuring around 1–2 cm, has a sweet-and-tangy flavor, and is usually sun-dried as a dried fruit after harvesting in late summer or early autumn (Figure 1). Although, the berry fruits from *L. barbarum* and its closely related species, *L. chinense, L. ruthenicum*, and *L. yunnanese*, are interchangeable, *L. barbarum* is the most dominant species and produces larger berry fruits compared to other species. *F. lycii* is traditionally used in Chinese home cooking in tea, soups, and porridge for its pleasant flavor and in the Chinese Pharmacopeia for about two centuries as an aid to vision and longevity as well as a remedy for diabetes to balance “*yin*” and “*yang*” in the body [3,4].

*F. lycii* is usually found in the arid and semi-arid regions of China and eastern Asia including Japan, Korea, and Vietnam. Currently, China is the world’s largest producer of *F. lycii* with farms in several regions, such as Qinghai, Xinjiang, Shaanxi, Gansu, Hebei, inner Mongolia, and Tibet. Approximately 90% of commercially available berry fruits are from the *L. barbarum* species in the autonomous regions of north-central China (Ningxia Hui) and western China (Xinjiang Uyghur), where they are grown according to Good Agricultural Practice [5]. *F. lycii* is fast gaining popularity as a functional food. As a result, cultivation of *F. lycii* has started in many regions in the west, including diverse parts of Europe and America, due to increasing demand. Besides being consumed as fresh or dried berry fruits, this ”Asian fruit” is also available in the form of processed food (cookies, chocolates, muesli, and sausages), and beverages (teas, fruit juices, wines, and beers) as well as herbal supplements that have been commercialized in western countries [6]. In recent years, several studies have been conducted to confirm and demonstrate its chemical ingredients, pharmacological properties, and clinical association with ocular health and diseases.

## 2. Bioactive Components

*F. lycii* is an excellent source of macronutrients, micronutrients, and several bioactive compounds distinguished by their high antioxidant potential (macronutrients: 46% carbohydrate, 13% protein, 1.5% fat, 16.5% dietary fiber; micronutrients: vitamins and minerals) [7]. Three bioactive compounds deserve special mention, and these are polysaccharides, carotenoids, and phenolics. The polysaccharide component (5–8% of dried fruit, hydrophilic) is the most important component and consists of six monosaccharides—glucose, galactose, arabinose, rhamnose, mannose, and xylose [8]. The carotenoids are the second most important component (0.03–0.5% of the dried weight, lipophilic) and are responsible for the characteristic bright and vivid orange to red coloration of these berries [9]. Flavonoids and phenolic acids constitute the phenolic component with quercetin-rhamno-di-hexoside and quercetin-3-O-rutinoside being the predominant flavonoids and chlorogenic acid being the most abundant phenolic acid [10]. Taurine, betaine (a natural amino acid), and heat stable vitamin C precursor represent the few other bioactive components that are found in high concentrations in these berries [11].

*F. lycii* is characterized by exceptionally high levels of the carotenoid zeaxanthin (Z) in the form of zeaxanthin dipalmitate (Figure 2) [12]. Fully ripe berry fruits contain around 60–70 times higher Z (mg/100 g fresh weight) when compared to egg yolk, another rich source of Z. This makes the berry fruits an outstanding source of Z in the human diet (*F. lycii* = 35.7 mg; egg yolk = 0.29 mg) [13]. The process of ripening of *F. lycii* is associated with two major changes: first, a change in the carotenoid profile from low levels of non-specific carotenoids to high levels of xanthophyll carotenoids, mainly esterified Z; second, a change in the carotenoid deposition state from chloroplast-specific solid state to chromoplast-specific liquid state. Ultrastructural study of *F. lycii* has shown that Z is deposited in a presumably J-aggregated liquid-crystalline state within the nanoscaled chromoplast tubules [14]. Although these berries constitute one of the richest sources of Z, corn, spinach, butternut squash, collard green, and tangerines represent a few other good sources of Z, which are usually consumed in the human diet.

It is important to note that the bioactive components of *F. lycii* may differ geographically as well as regionally within a country. For instance, Italian *F. lycii* has a higher total carotenoid and Z dipalmitate content when compared to its Chinese counterpart (Carotenoids in mg/100 g: Italian = 355.4, Chinese = 224.8; Z dipalmitate: Italian = 173.50, Chinese = 33.45) [15]. A relatively higher content of micronutrients, such as vitamin K, copper, selenium, and zinc, has also been found in Italian *F. lycii*. Although *F. lycii* originating from China are overall rich in all three bioactive components with higher total antioxidant activity, there are significant differences in the carotenoid content of *F. lycii* originating from four different regions within China (Gansu, Ningxia, Qinghai, and Xinjiang) [16]. *F. lycii* from Gansu region has the highest total carotenoid content (212.5–237.1 mg/100 dw), whereas the highest content of Z dipalmitate is found in *F. lycii* from Qinghai region (77.57–94.03 mg/100 dw). *F. lycii* from Xinjiang region has the lowest total carotenoid as well as Z dipalmitate content (carotenoid = 129.3 mg/100 dw, Z = 42.60 mg/100 dw). It is believed that differences in climate, nature of soil, cultivation methods, and post-harvesting processing methods (drying and storage conditions) may result in geographic and regional differences in the bioactive components of *F. lycii*.

## 3. Bioavailability of Bioactive Components

Carotenoids are one of the well-studied components of *F. lycii* in terms of bioavailability. Table 1 summarizes the studies that have evaluated the bioavailability of carotenoid component of *F. lycii*. *F. lycii* contains exceptionally high levels of Z dipalmitate, an esterified form of Z formed from a combination of Z and palmitic acid. During digestion, Z esters are hydrolyzed, resulting in free Z, which is then incorporated within the chylomicron component of triacylglycerol-rich lipoproteins and transported into the blood circulation [17]. Past studies have demonstrated a good degree of bioavailability of Z from *F. lycii* both in animals and humans. An increase in plasma as well as retinal levels of Z (two-fold increase in macular Z levels) was demonstrated in rhesus monkeys following six weeks of supplementation with dietary *F. lycii* extracts [18]. Similarly, around a 2.5-fold increase in plasma levels of Z was found in healthy adults following daily consumption of *F. lycii* for 28 days [19,20]. These observations suggest that a modest intake of *F. lycii* on a daily basis markedly increases plasma levels of Z. However, a relatively smaller increase in plasma Z levels was found in elderly subjects (mean age = 67 years) when compared to younger subjects (mean age = 28 years) following intake of these berries, suggesting that there may be an age-related decline in the absorption and/or uptake of Z. Of note, these studies involving human subjects did not measure the retinal levels of Z.

*F. lycii* not only represents the richest natural source of Z, but more importantly, the bioavailability of Z from this berry fruit is much higher than other dietary sources of carotenoids. The natural occurrence of the ester form of Z may account for the increased bioavailability of Z [24]. Experiments in rodents and humans showed enhanced bioavailability of esterified Z originating from *F. lycii* when compared to non-esterified Z [21,25], and this has been attributed to more effective micelle formation needed for lipase activity during the process of digestion. Alternatively, increased bioavailability of esterified Z from *F. lycii* may be due to the unique deposition state of Z. In *F. lycii*, Z exists in a liquid-crystalline state within tubular chromoplast (contrast solid protein-complexed Z within chloroplast in spinach) that may result in greater liberation and bioaccessibility of Z [13]. In addition, an increase in bioavailability of Z from *F. lycii* may also be due to the characteristic J-aggregates integral to the unique deposition state of Z. Esterification of hydroxyl group in Z dipalmitate results in development of loosely packed J-aggregates due to deterrence of hydrogen bond formation between molecules. An alternative type of aggregate for carotenoid Z is tightly packed H-aggregates that result from the formation of hydrogen bond between molecules. J-aggregated Z dipalmitate demonstrated higher bioavailability in human subjects (23% higher) as well as in-vitro bioaccessibility model (INFOGEST) when compared to H-aggregated Z [14].

The mode of consumption also affects the bioavailability of Z from *F. lycii*. Traditionally, *F. lycii* is often consumed in the form of infusion by adding hot water to the dried berries. Increase in infusion temperature and time is associated with increased activity of bioactive compounds, in particular antioxidant activity. However, *F. lycii* preparations of 100 °C for 1~3 h, 90 °C for 2~3 h, and 80 °C for 2.5~3 h were found to be equivalent in terms of antioxidant activity of bioactive compounds [23]. Additionally, homogenization of *F. lycii* in hot skimmed milk results in a formulation that has a three-fold enhanced bioavailability of Z compared with both the “classical” hot water and warm skimmed milk treatment of the berries [22]. Furthermore, higher processing temperature in presence of milk proteins further enhances Z bioavailability, probably by improving its incorporation into mixed micelles, uptake by enterocytes, and subsequent release in triacylglycerol-rich lipoproteins. Although both *F. lycii* and egg yolk are believed to be good dietary sources of Z, the former may represent a healthier choice, because intake of the latter is associated with incident cardiovascular diseases and all cause mortality in a dose-dependent manner [26].

Processing of *F. lycii* products may affect bioavailability of the bioactive components. In recent years, numerous processed products of *F. lycii* have become popular as ”super food” in the market, particularly in Europe and North America [27]. One such product is Himalayan Goji juice in the US that refers to the juice made from reconstituted extracts of four fruits including *F. lycii*, grape, apple, and pear. The process of heat treatment leads to degradation of carotenoids in *F. lycii* with an approximate loss of about 20–24% [28]. On the other hand, addition of citric acid can effectively control the degree of carotenoid oxidation and thereby improve the level of carotenoid, the optimal concentration being 0.2% [29]. Similarly, the process of fermentation for a period of 24 h is associated with a 17% reduction in carotenoid content and 87% decrease in sugar contents of these berries but has no effect on the polyphenol content [30]. Besides traditional extraction method using hot water, the bioactive components in *F. lycii*, particularly the polysaccharide component, can be extracted using novel extraction technologies, such as ultrasound assisted, enzyme assisted, microwave assisted, and supercritical fluid extraction methods. Yield of the polysaccharide component varies with the type of extraction method used with the microwave assisted method having the maximum yield of polysaccharide component (8.25% ± 0.07%) [31].

## 4. *Fructus lycii* and Retinal Diseases and Degeneration

This section comprehensively reviews the literature pertaining to *F. lycii* and three clinically important retinal conditions, namely age-related macular degeneration (AMD), diabetic retinopathy (DR), and retinitis pigmentosa (RP). Oxidative damage is one of the common underlying pathways for these retinal diseases. It refers to an imbalance between the production of reactive oxygen species (ROS) and the body’s ability to scavenge these ROS with its antioxidant defense system. Under physiological conditions, ROS regulates various biological functions by acting as a second messenger in cell signaling. Excessive production of ROS leads to oxidative damage to cellular structures directly by oxidizing lipids, proteins, and nucleic acids. The excess ROS also functions as signaling molecules to activate a number of cellular stress-sensitive pathways culminating in cellular damage [32]. The retina is particularly vulnerable to chronic oxidative damage due to several reasons [33,34]. First, photoreceptors and retinal pigment epithelium (RPE) have high metabolic activity resulting in generation of ROS as a natural by-product of mitochondria; second, photoreceptors are constantly exposed to light, which in the presence of high oxygen levels results in photooxidative damage; third, there is a high content of polyunsaturated fatty acids in the photoreceptor membranous disks; fourth, lipofuscin is derived, at least in part, from oxidatively damaged photoreceptor outer segments and that in itself is a photoreactive substance. Prolonged photooxidative damage eventually triggers other pathways such as retinal apoptosis, which adds to retinal damage.

*F. lycii* exhibits strong antioxidant action that is believed to be attributable to the synergistic effects of its bioactive components, carotenoids, polysaccharides, and flavonoids, each of which individually are strong antioxidants. Studies in animal models and humans have consistently demonstrated that *F. lycii* displays scavenging activity against superoxide anion and hydroxyl radicals, the two most important ROS, as well as inhibits lipid peroxidation of low density lipoprotein cholesterol fraction as evidenced by a decrease in malondialdehyde levels [35,36,37,38]. *F. lycii* also enhances endogenous antioxidant defense system by increasing the levels of antioxidant enzymes, such as superoxide dismutase, catalase, and glutathione peroxidase [39,40,41]. Moreover, administration of *F. lycii* in-vitro significantly modulates oxidative stress by inhibiting caspase-3 activation along with ROS levels in a concentration-dependent manner [42]. At the cellular level, *F. lycii* can enhance macrophage nitric oxide, phagocytic capacity, and acid phosphatase as well as down-regulate gene function to prevent ROS-induced apoptosis, thus resulting in significant increase in cell viability [43,44].

### 4.1. F. lycii and Age-Related Macular Degeneration

AMD is a degenerative disorder of the macula characterized by progressive loss of central vision due to an atrophic and/or neovascular event and is the leading cause of blindness among the elderly individuals over 50 years of age in developed countries [45]. In the coming years, this irreversible retinal disease will become a major public health burden globally due to rapidly ageing populations, particularly in Asia, which accounts for more than 60% of the world’s population [46]. Although the exact etiopathogenesis of AMD remains unclear, it is believed to be multifactorial in origin with existing scientific evidence suggesting oxidative damage as the main underlying mechanism.

Table 2 summarizes studies that have evaluated *F. lycii* with respect to AMD in human RPE cell lines, animal models, and humans. Pre-treatment with *F. lycii* extract protected human RPE cells against acute oxidative stress injury (apoptotic cell death) induced by exposure to blue light, ultraviolet light, and hydrogen peroxide [47,48,49,50]. Experimental studies have demonstrated visible morphological and functional changes in the RPE cells, such as an increase in the number of viable cells, reduction in apoptotic cells, enhanced proliferation, and phagocytic ability with consequential decrease in lipofuscin accumulation. The protective effect of *F. lycii* on RPE cells is believed to be exerted by the following mechanisms: first, antioxidant property resulting in decreased levels of endogenous ROS; second, modulating action on apoptosis-related genes leading to increase in Bcl-2/Bax ratio (up-regulation of Bcl-2, an anti-apoptotic gene and down-regulation of Bax, a pro-apoptotic gene) with resultant decrease in cell apoptosis; third, attenuation of deoxyribonucleic acid (DNA) damage in a dose-dependent manner. In-vitro studies evaluating the protective effect of *F. lycii* on RPE cells usually involve its aqueous fraction, and although both aqueous and ethanol extracts exhibit potent antioxidant activity, the ethanol extract exhibits a stronger antioxidant effect. This is probably attributable to the synergistic activity of hydrophilic (polysaccharide) and hydrophobic (carotenoids and flavonoids) bioactive components, both possessing strong antioxidant effects. Alternatively, the stronger antioxidant effect of ethanol extract may be due to the involvement of additional signaling pathways, such as toll like receptor pathways.

Similarly, in animal models, *F. lycii* effectively protected photoreceptor cells against light-induced retinal damage via its antioxidant properties [41,51]. In rats exposed to white light, there were decreased levels of oxidative stress markers, increased levels of endogenous antioxidants, and up-regulation of the antioxidative genes Nrf2 and TrxR1 following *F. lycii* administration. Moreover, a delay in apoptosis of photoreceptors was also observed in light-damaged retinas as suggested by decreased expression levels of PARP-14, a member of the poly-ADP-ribose polymerase family. DNA is particularly sensitive to light-induced oxidative damage, and PARP-14 plays a significant role in DNA repair. The process of apoptosis is associated with high levels of PARP due to DNA fragmentation related with apoptosis. Morphological changes, such as photoreceptor cell loss, nuclear condensation, an increased number of mitochondrial vacuoles, outer membrane disk swelling, and cristae fractures, as well as functional changes, such as loss of a- and b-wave amplitudes induced by light exposure, were also distinctively ameliorated by *F. lycii* in rats.

To date, there are three prospective randomized controlled clinical trials of *F. lycii* supplementation in human subjects with AMD. Subjects with early AMD demonstrated a delay in disease progression, as evidenced by an absence of macular hypopigmentation progression and soft drusen accumulation, following supplementation with *F. lycii* for a period of 90 days when compared to subjects in the control group [20]. This was accompanied by a 26% rise in plasma Z concentration and a 57% rise in antioxidant capacity in subjects on *F. lycii* supplementation, which is equivalent to 10 mg/day of Z. In the same cohort, *F. lycii* supplementation was also found to increase plasma antioxidant capacity along with enhanced immune defenses (higher immunoglobin G response, sero-conversion, and protection rates following flu vaccination) and improved *yin* deficiency [52]. In addition, supplementation with *F. lycii* for 90 days increased serum Z concentration, macular pigment optical density and visual function (best-corrected visual acuity) in patients with early AMD, without causing any detectable adverse effects [53]. The authors speculated that increase in macular pigment optical density in the central retina was the main reason for the improvement in visual function in early AMD. The main limitations of these studies are their relatively small sample size and short period of supplementation. Nevertheless, the findings are consistent with a protective effect of *F. lycii* in subjects with early AMD in terms of delaying disease progression and improvement in visual function. Although the underlying mechanism for the protective effect is believed to be accumulation of plasma Z and other antioxidants, no direct relationship was observed between changes in plasma Z and macular features.

The Age-Related Eye Disease Study 2 demonstrated a protective effect of lutein and Z supplementation on progression to advanced AMD when the subgroup analyses of the treatment effect were limited to those participants with the lowest dietary intake of lutein and Z [54]. *F. lycii* has exceptionally high level of carotenoids, particularly Z. Along with lutein, it selectively accumulates in the primate macula, around 500-fold higher concentration than in any other body tissues, where they are collectively referred to as macular pigment [55]. Although Z and lutein are structural isomers, there are a few important differences between these two carotenoids that may have functional implications: first, Z predominates in the central part of the macula, the fovea, whereas lutein predominates in the periphery, the ratio being 2.4:1 at the fovea [56]; second, Z contains two β-ring end groups, whereas lutein contains both a β-ring and a ε-ring [57]; third, Z is twice as efficacious as lutein in quenching ROS, and this is probably attributable to the extended conjugated system of Z [58]. Therefore, the potential ocular benefits of dietary consumption of *F. lycii* with its high Z content are self-explanatory.

### 4.2. F. lycii and Diabetic Retinopathy

DR is one of the most common microvascular complications of diabetes and remains a major cause of preventable blindness in the working-age population worldwide [59]. It has been estimated that more than one third of those with diabetes worldwide have some form of DR and nearly one in ten have vision threatening DR [60]. Although the molecular mechanisms underlying diabetic microvascular complications remain unclear, there is a growing body of evidence to suggest that hyperglycemia-induced oxidative stress and its downstream pathways play an important role in the pathogenesis of DR [61]. For instance, oxidative stress has been shown to induce the expression of pro- apoptotic molecules leading to apoptosis, which has been identified as one of the pivotal mechanisms for cellular damage in DR.

Table 3 summarizes studies that have evaluated *F. lycii* with respect to DR in animal models and human cell lines. To date, there is no study that has examined the relationship between dietary *F. lycii* and DR in human subjects. *F. lycii* administration has been shown to ameliorate retinal structural and functional changes due to diabetes in animal models [62,63,64]. Restoration of the overall retinal thickness and its individual layers, decreased structural disturbance of photoreceptors outer and inner segments, and decreased cavitation in the RPE cell layer represent just a few of the retinal structural changes observed. Similarly, structural changes altered in retinal vessels by *F. lycii* include reduction in basement membrane thickness, increase in vessel lumen, increase in morphologically normal capillaries, and reduction in abnormal vascular tuft and twisted capillaries suggestive of vascular proliferation. At the functional level, *F. lycii* administration was associated with blunting of diabetes-induced decrease in amplitude of a-wave, b-wave, and oscillatory potentials on electroretinography (ERG) in animal models of diabetes [65]. Moreover, there was also reversal of high glucose-induced Müller cell dysfunction and overexpression of glial fibrillary acid protein. Müller cells are the principal glial cells that provide nutritional and functional support to retinal neurons and exhibit the most significant changes of all the glial cells in DR [66]. The level of glial fibrillary protein is an established indicator of retinal stress and is marginally detectable in Müller cells under normal retinal conditions but markedly up-regulated in the diabetic retina [67]. Of note, b-wave is believed to reflect light-induced electrical activity in the bipolar cells with contributions by Müller cells; however, ERG observations are unable to differentiate diabetes-induced damage to bipolar cells and/or to the Müller cells.

Administration of *F. lycii* in diabetic rats was associated with reversal of diabetes-induced increase in vascular endothelial growth factor (VEGF) levels and suppression of pigment epithelium derived factor (PEDF) levels [65]. This resulted in reinstallation of the balance between angiogenic and anti-angiogenic factors and decreased likelihood of angiogenesis. There was also significant reduction in the levels of VEGF mRNA in the retina of experimental animals treated with *F. lycii*. VEGF is a pro-inflammatory peptide that is necessary for the normal maintenance of retinal and choroidal vessels, but elevated expression of VEGF induced by hypoxia is a key stimulus for aberrant growth of new vessels in proliferative DR [75]. On the other hand, PEDF is one of the protective factors that promotes anti-angiogenesis and counteracts the pro-inflammatory environment in DR. Proliferative DR is one of the main vision-threatening complications of diabetes and is characterized by pathological neovascularization of the retina [76].

*F. lycii* may exhibit protective effect on the outer as well as inner blood–retinal barrier (BRB) as suggested by experimental studies [69,71,74]. Diabetes-induced dysfunction of the BRB results in leakage of fluid and circulating proteins within the neural retina and this is known as diabetic macular edema. Currently, diabetic macular edema is the most prevalent vision-threatening complication of diabetes [77]. The outer BRB is formed by tight junctions between the RPE cells, a monolayer of epithelial cells that separate the vascular choroidal system from the neurosensory retina. Diabetes-induced hyperglycemia increases intracellular glucose in RPE cells, which stimulates cytosolic isoform of adenylyl cyclase, resulting in increase in intracellular cyclic AMP levels and functional impairment of the outer BRB. High glucose-induced RPE barrier impairment is ameliorated by *F. lycii* by reversing the activity of adenylyl cyclase and decreasing cytosolic AMP levels [74]. It has been hypothesized that this action of *F. lycii* is mediated via its taurine bioactive component interacting with the catechol-estrogen binding site of soluble adenylyl cyclase. *F. lycii* may also protect the integrity of the inner BRB, formed by tight junctions between retinal vascular endothelial cells, by targeting the ROCK signaling pathway. The ROCK pathway regulates cellular adherence, migration, proliferation, and apoptosis through the control of the actin cytoskeletal assembly and cell contraction [78,79]. In particular, this pathway regulates expression and function of intercellular adhesion molecule-1 in endothelial cells as well as directly phosphorylates occludin and other tight junctional proteins. It is believed that the retinal vascular endothelial cells are in ROCK-activated state. *F. lycii* administration in diabetic rats was associated with reversal of hyperglycaemia-induced up-regulation of ROCK pathway and down-regulation of phosphorylated occludin with subsequent VEGF induced hyper permeability and vascular leakage [71].

*F. lycii* may attenuate hyperglycemia-induced apoptosis of RPE cells through up-regulation of PPAR-γ and down-regulation of caspase-3 pathway. PPAR is prominently expressed in photoreceptor outer segments, RPE cells, and choriocapillaris in mammalian eyes, and ligand-activated PPAR-γ controls apoptosis induced by oxidative stress and thus contributes to retinal protection. Retinal expression of PPAR-γ is suppressed in experimental models of diabetes and in endothelial cells exposed to high glucose [80,81]. *F. lycii* extract enhanced expression of PPAR-γ-related mRNA and protein in a dose-dependent manner in human RPE cell lines exposed to high glucose [73]. At the same time, expression as well as enzymatic activity of caspase-3 was down-regulated [82,83]. Members of the caspase family are involved in the initiation and execution of apoptosis, and caspase-3, known as the executioner caspase, plays an important role in the proteolytic cascade during apoptosis. The cytoprotective effect of *F. lycii* on RPE cells, which is regulated via PPAR-γ-mediated caspase-3 pathway, is believed to be due to its bioactive component taurine [72]. Several lines of evidence suggest that taurine in *F. lycii* plays an important role in DR prevention. Taurine is a non-essential amino acid found abundantly in the retina and has an ability to cross the BRB [84]. In fact, the taurine concentration in the retina including photoreceptors and RPE cells is estimated to be around 60–80 mM, corresponding to 40–75% of the total free amino acid content deemed necessary to maintain physiological functions, such as membrane stabilization, neuromodulation, and maintaining the integrity of the retina [85,86]. Past studies have shown that plasma and tissues levels of taurine are reduced in diabetes, and there is a beneficial effect of taurine supplementation on prevention and amelioration of hyperglycemia-induced retinal changes, such as retinal glial cell apoptosis [87,88,89]. However, more conclusive studies in human subjects are warranted to understand the mechanism of action of taurine and its long-term safety and efficacy in DR.

At the cellular level, *F. lycii* may reduce oxidative stress to mitochondria and endoplasmic reticulum (ER), the two prime targets for diabetes-induced oxidative damage. Mitochondria, the powerhouse of the cell, are the major endogenous source of ROS but are also the targets for its damaging effects. Sustained exposure to ROS (oxidative stress) damages the mitochondria and compromises the electron transport system, which ultimately results in mitochondrial DNA damage with subsequent impairment of transcription, and propagation of a vicious cycle of ROS generation sets in [90]. Indeed, DNA damage and apoptosis of mitochondria has been associated with changes in retinal blood flow and breakdown of BRB in late stages of DR [91]. Experimental studies have demonstrated that *F. lycii* may enhance mitochondrial biogenesis via up-regulation of carotenoid metabolic genes in diabetic retina [68]. Diabetes-induced hyperglycemia and subsequent hypoxia initiate changes in carotenoid homeostasis through inhibition of metabolic gene expression, which in turn leads to inhibition of 5′ AMP-activated protein kinase (AMPK), decrease in mitochondrial transcription factor (TFAM), and mitochondrial dysfunction [92,93]. Administration of *F. lycii* in diabetic mice primarily activated AMPK alpha 2 in mitochondria and nuclei, which in turn triggered increased expression of genes related to metabolic homeostasis of carotenoid, particularly scavenger receptor B1 (SR-B1), glutathione S-transferase pi 1 (GSTP1), and beta-carotene oxygenase 2 (BCO2). In tandem, there was increased expression of proteins responsible for mitochondrial biogenesis (peroxisome proliferator-activated receptor γ coactivator-1 [PGC-1-alpha], nuclear respiratory factor 1 [NRF1], and TFAM) and decreased expression of proteins produced in response to cellular stress (hypoxia inducible factor 1-alpha [HIF-1-alpha], VEGF, and heat shock protein 60 [HSP60]). As a result, there was enhancement of mitochondrial function and subsequent neuroprotection in diabetic mice. Similarly, *F. lycii* may also attenuate ER stress as demonstrated by decreased expression of ER stress biomarkers in diabetic retina under experimental settings [70]. Hyperglycaemia-linked oxidative stress disrupts protein synthesis and protein folding within the ER that eventually leads to ER stress and subsequent cell apoptosis. *F. lycii* restored AMPK and its downstream target proteins (retinal FOX03α essential for cell survival) in diabetic retina and in human RPE cells exposed to high glucose levels [62]. Consequentially, there was increased expression of antioxidant enzymes (thioredoxin and superoxide dismutase) with resultant normalization of cellular ROS status and redox homeostasis as well as decreased ER stress. This study also documented that restoration of AMPK by *F. lycii* was mediated, at least partially, through its bioactive carotenoid component Z.

### 4.3. F. lycii and Retinitis Pigmentosa

RP is a heterogeneous group of inherited bilateral retinal pigmentary dystrophies that are characterized by progressive and sequential loss of rod and cone photoreceptors, ultimately leading to complete blindness [94]. Several mechanisms have been linked to triggering the death of photoreceptors in mouse models of RP, such as oxidative stress, ER stress, dysregulation of cyclic guanosine monophosphate (cGMP) signalling, accumulation of calcium ions, and inflammatory responses [95]. Oxidative stress may be a crucial pathway because of the following reasons: first, the retina is highly vulnerable to oxidative stress; second, antioxidant treatment improves cell survival and preserves photoreceptors function in animal models; third, antioxidant treatment may decrease inflammatory markers involved in apoptosis [96]. Although experimental therapies under investigation are aiming to repair or rescue impaired vision, an effective treatment has yet to be discovered [97,98,99]. Currently, neuroprotection using antioxidant is widely employed as a therapeutic approach to delay photoreceptor degeneration in RP patients [100].

Table 4 summarizes studies that have evaluated *F. lycii* with respect to RP in animal models as well as human subjects. *F. lycii* demonstrated a protective effect on the structure and function of retinal neural cells via its strong antioxidant property in studies involving fast degenerating photoreceptor mouse model mimicking RP [101,102]. There was restoration of photoreceptor and bipolar cell morphology, improvement in arrangement of photoreceptor cell layer, maintenance of ramified shape of microglial cells, and preservation of the outer retinal thickness in the central retina. In the retinal ganglion cells, there was an increase in response to saturated light intensities, enhanced light evoked responses, increase sensitivity, and response speed along with a decrease in spontaneous abnormal firing. There was also differential effect with ON responses better than OFF responses at early stages of photoreceptor degeneration and enhanced OFF responses than ON responses at later stages. Functionally, photopic b-wave changes in the form of decreased latency and increased amplitude as well as larger scotopic a- and b- waves were observed on ERG. These findings suggest that *F. lycii* may potentially improve visual processing at multiple stages of information transmission via protecting photoreceptor, bipolar cells, and ganglion cells, and perhaps, function of higher brain centers, as suggested by enhanced visual behavior.

Furthermore, *F. lycii* may exhibit a long term protective effect on photoreceptors via multiple pathways in experimental settings. Following *F. lycii* administration, an increase in antioxidant activity was observed in animal models, as indicated by higher glutathione redox/antioxidant ratio, a ratio that is commonly used to measure oxidative stress status [102]. *F. lycii* also demonstrated an anti-apoptotic effect by attenuating photoreceptor cell apoptosis through modulation of PARP and caspases levels. Administration of *F. lycii* in animal models was associated with significant down-regulation of 9, 3, 7 pro, and cleaved caspases along with increased nuclear levels of PARP as well as cleaved PARP protein [103]. PARP is one of the substrates for caspases and activated caspase 3/7 cleaves PARP to elicit apoptosis. Additionally, *F. lycii* regulated photoreceptor cell apoptosis, at least partly, through down-regulation of HIF-1α and Bax protein expression in animal models [102]; the former regulates many biological processes, one of these processes being activation of transcriptional activity of p53, resulting in transcription of many pro-apoptotic proteins, such as Bax and caspase-3. In addition, *F. lycii* mediated expression of inflammatory mediators in animal models, partially through the nuclear factor kappa-light-chain enhancer of the activated B cells (NF-kB) signaling pathway, which is probably attributable to its polysaccharide bioactive component [102].

Similar to experimental studies in animal models, *F. lycii* may exhibit neuroprotective effect on the human retina and might help to minimize or delay cone degeneration in RP patients [104]. In a double-blind placebo-controlled trial with 42 RP patients, there was preservation of visual acuity, both high (90%) and low (10%) contrast, as well as average macular thickness following oral administration of *F. lycii* for a period of 12 months, but no significant effect was observed on visual field sensitivity or on any of the parameters of ERG. The main limitations of this study were its relatively small sample size, short duration of study period, and the active ingredients in *F. lycii* granules were not measured precisely. In addition, although the primary end-points (visual acuity and macular retinal thickness) reached statistical significance, these parameters may not be clinically significant. The authors proposed that *F. lycii* may be useful as a dietary supplement to effectively preserve photopic vision in RP patients and to help maintain quality of life; however, a large-scale longitudinal study is warranted to further investigate the long-term protective effect of *F. lycii* in RP patients with different genotypes.

## 5. Limitations of the Existing Studies

The majority of studies that have provided vital information on *F. lycii* in AMD, DR, and RP have been conducted in animal models or human cell lines. Experimental studies involving animal models are overall relatively inexpensive, easy to perform, and replicable; however, there are several limitations of these studies: (1) there are significant variances between humans and animals due to differences attributable to species. For instance, rodents are characterized by the absence of a macula, and rod cells are the predominant retinal photoreceptors, whereas humans have specialized macula with cone cells as the main photoreceptors [105]; (2) nearly all studies have very small sample sizes without evidence of power calculations, thereby making it difficult to draw impactful conclusions; (3) the dosages used in animal studies are usually much higher than those used clinically in humans; (4) it is difficult to measure carotenoid levels in the mouse retina, thereby hindering the evaluation of *F. lycii* function (due to bioactive component carotenoid) in retinal protection and diseases; (5) it remains unclear whether the histological and biochemical changes observed under experimental settings can be translated to prevent and/or delay retinal diseases in real world clinical settings.

As for DR, no single diabetic mouse model recapitulates all the features or complications of human diabetes. Generally, streptozotocin-induced type I diabetic rat models manifesting the characteristic of hyperglycemia and impaired retinal function were used for evaluating disease processes associated with DR. On the other hand, genetically programmed diabetic db/db mouse models were used for studying the development and progression of DR in type 2 diabetes. The young diabetic db/db mouse with age less than 18 weeks do not exhibit signs of DR and thus represent good animal model for studying pathogenesis of DR in type 2 diabetes. Experimental studies in AMD exposed human RPE cell lines to stimulus, such as hydrogen peroxide, blue light irradiation, ultraviolet-B light, and high intensity white light, creating acute oxidative injury, thus imitating photooxidative injury in AMD. Of note, AMD appears to involve complex interaction of genetic and environmental factors (such as chronic photooxidative damage), making it a difficult disease to mirror in animal models. Fast degenerating photoreceptors models, such as wild rd 1 and rd 10 mice, or N-methyl-N-nitrosourea-induced photoreceptor apoptosis in Sprague–Dawley rats, were used to mimic RP in experimental settings. Nevertheless, promising findings identified from experimental studies should warrant greater interest in this field of research.

## 6. Side Effects and Drug Interaction

*F. lycii* is considered a safe dietary supplement and a dose of 15 g per day, equivalent to 3 mg/day of Z, is deemed beneficial to ocular health [106]. In order to reap the full health benefits of *F. lycii*, it is important to ensure that these berry fruits are free of any external contamination that may occur during the process of production and/or post marketing. A small amount of atropine alkaloid exists as a natural content in *F. lycii* with a maximum detectable concentration of 19 parts per billion, which is believed to be far below the toxic levels in humans [107]. To date, none of the published reports have described any case of atropine-related side effects following *F. lycii* intake.

Although allergic reactions to *F. lycii* are extremely rare, a few cases have been reported in the literature with symptoms, such as generalized urticaria, toxic hepatitis, and rare life-threatening anaphylactic reaction [108,109]. It is believed that lipid transfer proteins may be involved in allergic sensitization of *F. lycii*, and therefore there is a high degree of cross-reactivity of *F. lycii* with peach peel and tomatoes. Caution should also be exercised in individuals with a history of any food allergy or pan allergy with nonspecific lipid transfer proteins [110,111]. In addition, there are reports describing spontaneous hemorrhages in patients on warfarin who consumed tea, juice, and wine containing *F. lycii*. It is believed that these hemorrhages are attributable to increased international normalized ratio, possibly due to interaction of *F. lycii* with warfarin, an anticoagulant medicine [112,113,114]. A history of *F. lycii* intake should be made mandatory in individuals who take medications with a narrow therapeutic index.

## 7. Future Directions and Conclusions

It is evident that existing literature in relation to *F. lycii* and retinal diseases detailed herein are mainly from experimental studies in animal models or human cell lines with a paucity of data from human subjects. Nonetheless, promising observations from these studies suggest that *F. lycii* has the potential to prevent and/or delay progression of retinal diseases (Figure 3) along with a favorable safety profile with few, if any, concerns regarding its adverse reactions.

However, there still remain a number of concerns that need to be addressed before *F. lycii* establishes its place in the clinical management of retinal diseases. First and foremost, observational and interventional studies, particularly long-term supplementation studies using *F. lycii* involving human subjects with retinal diseases, are warranted in the future to confirm the observed protective effect of *F. lycii* in retinal diseases. Second, there is no research grade *F. lycii* currently available in the market for conducting scientific studies, resulting in lack of standardization and quality control for the bioactive components in *F. lycii* along with adverse effects of post-marketing surveillance as another area of concern. It is thus crucial to ensure the availability of research grade *F. lycii* with consistent concentration of bioactive components to promote quality research in this field. Third, of all the bioactive components, carotenoids are the only bioactive component that can be precisely measured in terms of dietary intake and plasma levels as well as tissue levels, i.e., in the retina, that corresponds to the anatomical site of retinal diseases. Given that the biological effects of *F. lycii* are probably attributable to the additive, complementary, and/or synergistic effects from multiple bioactive components, one of the biggest challenges for future research is to segregate the specific properties of the various other bioactive components. Fourth, future studies should also examine whether the retinal benefits of *F. lycii* are influenced by additive or synergistic interactions with phytochemicals from other dietary sources typically consumed in the human diet. Fifth, research efforts should also be focused on studying the role of nutrigenomics (effects of nutrients on the genome, proteome, and metabolome) as well as nutrigenetics (effects of genetic variation on the interaction between diet and disease) in relation to *F. lycii*. Lastly, at the basic science level, studies should focus on the high order structures of bioactive components and relationship between structure and bioavailability to enhance our understanding and knowledge of *F. lycii*.

In summary, *F. lycii* offers an inexpensive and relatively safe “whole food” dietary supplement for the maintenance of retinal health as well as for prevention and/or delay in progression of retinal diseases commonly seen in clinical practice. It is important to note that these exotic berry fruits must be retailed with strict adherence to pharmacopoeia-recommended guidelines, including dosage regimes. In the near future, *F. lycii* supplementation may represent a model for modern medicine with successful integration between traditional Asian and Western medicine reaping the advantage of multi-functional herbal ingredients in the prevention and treatment of retinal diseases. We hope that this review article will not only increase awareness amongst the medical and scientific community regarding published work on *F. lycii* and retinal diseases, but will also help promote robust research work in this area to establish dietary intervention strategies aiming to promote ocular health and prevent common retinal diseases in the ageing population.

## Figures and Tables

**Figure 1 nutrients-13-00246-f001:**
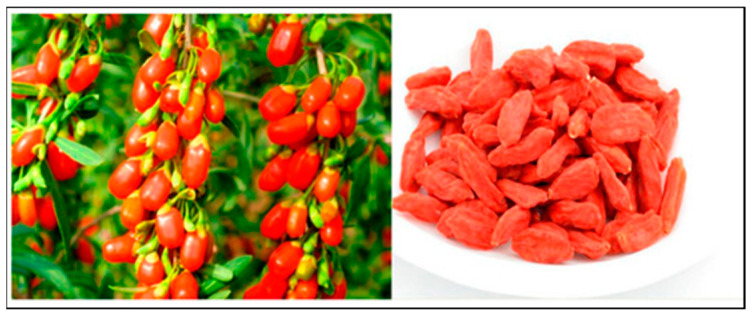
The fresh (**left**) and dried (**right**) berry fruits of *Lycium barbarum* ‘*Fructus lycii*’.

**Figure 2 nutrients-13-00246-f002:**
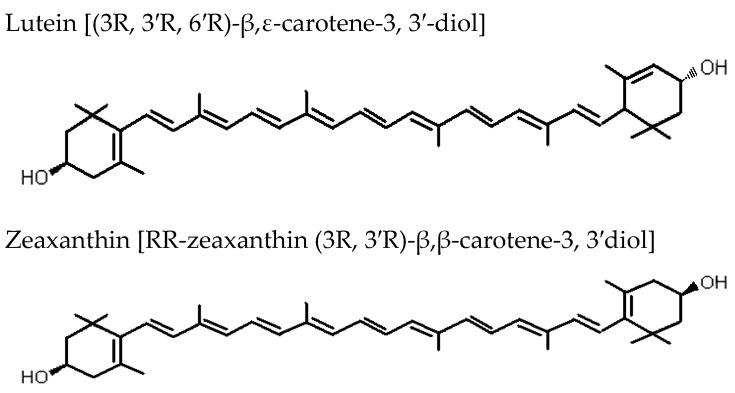
Biochemical structures of the carotenoids lutein and zeaxanthin.

**Figure 3 nutrients-13-00246-f003:**
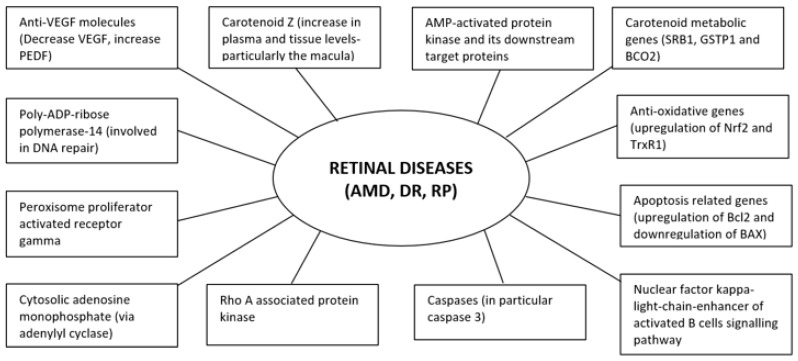
Molecular targets of *F. lycii* in amelioration of retinal diseases.

**Table 1 nutrients-13-00246-t001:** List of studies examining the bioavailability of carotenoid zeaxanthin in *Fructus lycii*.

Authors (year)	Type of Study	Subjects	*F. lycii* Formulation	Aim	Conclusion
**HUMAN STUDIES (HEALTHY SUBJECTS)**					
Breithaupt et al. (2004) [21]	Single-blinded, crossover study (3-week of depletion period)	*N* = 12, Males = 6; Females = 6	A Z-standardized dose (5 mg) suspended in yoghurt, esterified (3R,3R’-Z dipalmitate from *F. lycii*) and non-esterified 3R,3R’-Z forms	Bioavailability of esterified Z versus non-esterified Z	Administration of both esterified and non-esterified Z increased plasma Z levels; however, levels were significantly higher with former suggesting an enhanced bioavailability of esterified form of Z.
Cheng et al. (2005) [19]	Single-blinded, placebo-controlled study	*N* = 27, supplementation group = 14; age and gender-matched controls = 13	Whole *F. lycii* (15g/d, equivalent to about 3mg Z) for 28 days	Change in fasting plasma Z levels	Following supplementation, there was 2.5-fold rise in plasma Z levels suggesting that Z in whole *F. lycii* was bioavailable and that modest daily intake will markedly increases fasting plasma Z levels.
Benzie et al. (2006) [22]	Double-blinded, crossover study (3-5 week wash out period)	*N* = 12, Males = 5, Females = 7	A Z-standardized dose (15 mg) from freeze dried powder of *F. lycii* homogenized in 3 forms: warm skimmed milk (40°); hot skimmed milk (80°); hot water (80°, control)	Bioavailability of different formulations of *F. lycii*	Homogenization of *F. lycii* in hot skimmed milk resulted in a formulation that has a 3-fold enhanced bioavailability of Z compared with both the ‘classical’ hot water and warm skimmed milk treatment of berries.
Bucheli et al. (2011) [20]	Randomized, double masked, placebo controlled study	*N* = 150, supplementation group = 75, placebo group = 75	Milk based formulation of *F. lycii* (equivalent to 10mg/d Z) for 90 days	Effect on plasma Z levels	Daily dietary supplementation of *F. lycii* increased plasma Z levels by 26% and plasma antioxidant capacity by 57% in elderly subjects aged 65–70 years.
Hempel et al. (2017) [14]	Randomized, single-blinded, two-way crossover study	*N* = 16, Males =8, Females = 8	Z extracted from *F. lycii* in two forms: H-aggregated Z and J-aggregated Z (equivalent to 10 mg Z)	Bioavailability of H aggregated versus J-aggregated Z in vivo (humans) and in-vitro (INFOGEST digestion protocol)	Overall, J-aggregated Z showed marginally higher bioavailability (23%) than H-aggregated Z in both humans and in-vitro models but bioaccessibility (micellization rate) were seen higher with H-aggregated Z. Combined effect of aggregation and esterification represents a small magnitude of effect on Z bioavailability in humans.
**IN-VITRO STUDIES**					
Hempel et al. (2017) [13]	In-vitro study	In-vitro digestion model	Fresh (both unripe and fully ripe) and dried berry fruits of *F. lycii*	Ultrastructure of *F. lycii* during fruit ripening; carotenoid profiling of unripe and ripe berry fruits; in-vitro bioaccessibility of xanthophyll carotenoids in dried *F. lycii* versus fresh spinach	There was ripening induced modification in carotenoid profile (high amounts of xanthophyll esters, mainly Z dipalmitate) and deposition (tubular chromoplasts presumably containing liquid-crystalline state of Z). *F. lycii* might represent a more potent source of xanthophyll carotenoid than green leafy vegetable due to enhanced liberation and bioaccessibility of Z from these berries.
Sun et al. (2017) [23]	In-vitro study	*F. lycii* Infusion	5 g of *F. lycii* in 150 mL of *F. lycii* infusion in hot water at different temperatures (60 °C, 70 °C, 80 °C, 90 °C, 100 °C), for different lengths of time (15, 30, 60, 120, 150, 180 min), and for different infusion times (1, 2, 3, 4 times)	Assessment of bioactive compounds and antioxidant activity of *F. lycii* infusions	The bioactive compounds and antioxidant activity of *F. lycii* infusion increased with increased infusion temperature and time but were equivalent with preparation conditions of 100 °C for 1~3 h, 90 °C for 2~3 h and 80 °C for 2.5~3 h. The antioxidant activity was contributed by polyphenols followed by flavonoids, carotenoids and polysaccharides.

**Table 2 nutrients-13-00246-t002:** List of studies that have evaluated *F. lycii* and age-related macular degeneration.

SNo	Authors (year)	Study Subjects	Administration of *F. lycii*	Impact on Variables	Conclusion
**A**	**STUDIES IN ANIMAL MODELS/CELL LINES**				
1	Dong et al. (2013) [47]	Human RPE cells (blue light)	Pretreatment with *F. lycii* in 3 concentrations (0.01 mg/mL, 0.1 mg/mL, 1 mg/mL)	Decreased levels of ROS and apoptotic cells	*F. lycii* pre-treatment can protect RPE cells against blue light induced damage via inhibiting ROS over generation and subsequent apoptosis in RPE cells.
2	Du et al. (2013) [48]	Human RPE cells and porcine photoreceptor outer segments	*F. lycii* in culture medium in 3 concentrations (0.01 mg/mL, 0.1 mg/mL, 1 mg/mL)	(1) RPE cells—increased proliferation and ability to phagocytose photoreceptor outer segments (2) decreased photoreceptor outer segment-induced lipofuscin accumulation in RPE cells	*F. lycii* treatment enhanced the ability of RPE cells to phagocytose photoreceptor outer segments along with proliferation and removal of lipofuscin.
3	Liu et al. (2015) [49]	Human RPE cell line (exposed to H_2_O_2_)	Pretreatment with different concentrations of *F. lycii* (0, 10, 50, 100, 500, 1000, 5000 ug/mL)	(1) Prevented loss of cell viability with maximal effect at 500 ug/mL (2) Reduced cell apoptosis (3) Inhibited up-regulation of Bcl-2 and down-regulation of Bax gene	*F. lycii* protected RPE cells against H2O2-induced acute oxidative stress injury (apoptotic cell death). This is probably attributable to increase in Bcl-2/Bax due to up-regulation of Bcl-2 and down-regulation of Bax expression.
4	Hsieh et al. (2018) [50]	Human RPE cell line (exposed to UVB)	Pretreatment with aqueous and ethanol extracts of *F. lycii* (from 0–200 ug/mL for 2 h)	(1) Prevented loss of cell viability (2) Reduced endogenous ROS levels (3) Reduced cell apoptosis (4) Attenuated loss of mitochondrial membrane potential (5) Dose dependent protection against DNA damage as measured by γH2AX levels (6) Prevented G2/M-arrest	*F. lycii* demonstrated protective effect on oxidative-induced apoptosis of RPE cells exerted via antioxidant property and protective activity on growth arrest as well as DNA damage. Ethanol extract showed stronger antioxidant effect when compared to aqueous extract.
5	Cheng et al. (2018) [41]	Rats, Male Sprague-Dawley (exposed to white light)	Diet supplemented with *F. lycii* 250 mg/kg for 54 days, submicron (particle size = 100 ± 70 nm) or blended (particle size = 3.58 ± 3.8 µm) type	(1) Maintained outer nuclear layer thickness (2) Preserved a and b waves on ERG, decreased MDA levels, higher total glutathione levels	*F. lycii* has protective effect on light induced retinal degeneration via its antioxidant property as evident by lower MDA and higher total glutathione levels. Submicron type provided better protection than blended type probably due to improvement in pharmacokinetics.
6	Tang et al. (2018) [51]	Mice, BALB/cJ (exposed to white light)	*F. lycii* (150 mg/kg, low dose, 300 mg/kg, high dose) once per day for 7 days	(1) Attenuated cell nuclei loss in outer nuclear layer (2) Ameliorated light induced damage in the form of decrease ROS production and increased rhodopsin (3) Prevented decrease in a- and b- wave amplitudes (6) Increased mRNA levels of TrxR1 and Nrf2 and decreased mRNA levels of PARP14	*F. lycii* protected photoreceptor cells against light-induced retinal damage probably due to decrease ROS production and up-regulation of anti-oxidative genes (Nrf2 and TrxR1). The resultant decrease in oxidative stress leads to reduction in mitochondrial damage and in apoptosis of photoreceptors.
**B**	**STUDIES IN HUMAN SUBJECTS**				
7	Bucheli et al. (2011) [20]	150 healthy subjects (65–70 years)	*F. lycii* (milk based formulation, 13.7 g/d) for 90 days, Placebo-controlled, double-masked, randomized	Subjects with *F. lycii* supplementation demonstrated: (1) Decreased hypopigmentation and soft drusen accumulation in macula (2) Increased plasma Z levels by 26% (3) Increased antioxidant capacity by 57%	*F. lycii* supplementation was associated with prevention of early AMD features, such as macular hypopigmentation and soft drusen accumulation, due to its antioxidant activity.
8	Vidal et al. (2014) [52]	150 healthy subjects (65–70 years)	*F. lycii* (milk based formulation, 13.7 g/d) for 90 days, Placebo-controlled, double-masked, randomized	Subjects with *F. lycii* supplementation demonstrated: (1) Increased plasma anti-oxidant capacity (2) Higher IgG antibody response, sero-conversion and protection rates following influenza vaccine (3) Improved syndrome of Yin deficiency	*F. lycii* supplementation reinforced immune defenses in elderly subjects, probably attributable to its antioxidant property, thus decreasing the likelihood of developing AMD.
9	Li et al. (2018) [53]	114 subjects with AMD (51 to 92 years)	*F. lycii* supplementation (25 g/day for 90 days) Prospective, randomized controlled trial	AMD subjects with *F. lycii* supplementation demonstrated (1) Three-fold increased serum Z but not lutein (2) Increased macular pigment optical density (3) significant increased best corrected visual acuity	*F. lycii* supplementation increased serum Z, macular pigment as well as visual function (visual acuity) in patients with early AMD without causing any detectable adverse effects.

RPE: Retinal pigment epithelium; ROS: Reactive oxygen species; POS: photoreceptor outer segment; ARPE-19: Arising retinal pigment epithelial cell line-19; H2O2: Hydrogen peroxide; Bcl-2: B-cell lymphoma 2; UVB: ultraviolet B-rays; MDA: Malondialdehyde; ERG: Electroretinogram; TrxR1: Thioredoxin reductase 1; Nrf2: Nuclear factor (erythroid-derived 2)-like 2; PARP 14: Poly (adp-ribose) polymerase member 14: mRNA: messenger RNA.

**Table 3 nutrients-13-00246-t003:** List of studies that have evaluated *F. lycii* with respect to diabetic retinopathy.

SNo	Authors (year)	Experimental Model	*F. lycii* Administration	Impact on Study Variables	Conclusion
**A ANIMAL MODELS**					
1	Tang et al. (2011) [62]	Mice (Spontaneous diabetes)	*F. lycii* diet 1% (Kcal), duration = 8 weeks	Structural: maintained thickness of whole retina and integrity of RPE and ganglion cells Biochemical: decreased ER stress biomarkers-BiP, PERK, ATF6 and caspase 12, restored AMPK & FOX03α; increased antioxidant enzymes (thioredoxin & MnSOD)	*F. lycii* ameliorated retinal structure abnormalities at early stage of diabetes by mitigating cellular oxidative stress and/or ER stress, at least partly due to carotenoid component Z.
2	Hu et al. (2012) [63]	Rats (Streptozotocin induced DM)	*F. lycii* (Dried powder decocted in distilled water) 5 g/kg/d orally by gavage, duration = 8 weeks	Structural: maintained thickness of retina, absent abnormal vascular tufts and cavities in photoreceptor segment Functional: reversal of reduction in a- and b-wave amplitude	*F. lycii* exhibited protective effect against degenerative and apoptotic changes in the retina as well as loss of retinal function secondary to diabetes.
3	Guo et al. (2013) [64]	Rats (Streptozotocin induced DM)	*F. lycii* (gastrogavage), duration = 24 weeks	Structural: decreased pathological changes in mitochondria, reduced neural cell apoptosis and cell degeneration (changes limited to inner nuclear layer if present)	*F. lycii* decreased pathological changes in mitochondria as well as apoptosis of neurons and associated glial cells in diabetic retina.
4	Yu et al. (2013) [68]	Mice (Spontaneous diabetes)	*F. lycii* diet, duration = 8 weeks	Structural (RPE): Ameliorated mitochondrial dispersion along with relocation of mitochondria, increased pigment granules Biochemical: Activated AMPK-α2 in mitochondria and nuclei, increased expression of carotenoid metabolic genes (SRB1, GSTP1, BCO2), mitochondrial biogenesis (PGC-1α, NRF1 &TFAM), decreased cell stress responses (HIF-1α &VEGF -hypoxia related; HSP60-mitochondrial stress related).	*F. lycii* up-regulated carotenoid metabolic genes (via activation of AMPK-α2) along with enhanced mitochondrial biogenesis and reduced cellular stress responses. This resulted in reversal of mitochondrial dysfunction with consequential neuroprotection of the diabetic retina.
5	Li et al. (2014) [69]	Rats (Streptozotocin induced DM)	*F. lycii* (Intragastric administration, 250 mg/kg)	Structural: Decreased abnormality in shape and diameter as well leakage of retinal vessels, as suggested by lowered Evan blue content in retina Biochemical: decreased levels of VEGF expression.	*F. lycii* alleviated DM-induced retinal vasculopathy via protection of blood retinal barrier resulting in decreased vascular leakage of diabetic retina.
6	Wang et al. (2017) [70]	Rats (Streptozotocin induced DM)	*F. lycii* (Intragastric administration, 0.5 mL 6% once a day), duration = 24 weeks, Placebo-controlled	Structural: Prevented pathological changes in photoreceptors and ganglion cells, reduced mitochondrial changes in bipolar and Müller cells (shorter and reduced cristae) Biochemical: higher SOD activity, reduced MDA expression level, reduced VEGF mRNA and protein levels.	*F. lycii* alleviated DM-induced retinal neuropathy by reducing oxidative damage to mitochondrial pathway (particularly in bipolar and Muller cells) through its anti-oxidative effect. Consequently, there was reduction in apoptosis of neural tissue in diabetic retina.
7	Yao et al. (2018) [65]	Rats (Streptozotocin induced DM)	*F. lycii* (oral administration of 400 mg/kg/d or 200 mg/kg/d), duration = 20 weeks	Structural: increased retinal thickness, less disorganization of photoreceptor segments, reduced numbers of pyknotic nuclei, morphologically normal retinal capillaries (reduced thickness of basement membrane, increased vessel lumen, close attachment to endothelial cells) Functional: blunting of decreased amplitude of a-wave, b-wave, and oscillatory potentials, improved blood flow in CRV Biochemical: reversed of diabetes induced elevation of VEGF and GFAP and suppression of PEDF.	*F. lycii* protected retinal function and morphology of diabetic retina probably through reinstallation of the balance between angiogenic (GFAP and VEGF) and anti-angiogenic (PEDF) factors indicating that *F. lycii* may be a potential therapeutic agent for DR.
8	Wang et al. (2019) [71]	Rats (Streptozotocin induced DM)	*F. lycii* (250 mg/kg/day by oral gavage), duration = 12 weeks	Structural: prevented changes in overall retinal thickness, decreased structure disturbance of photoreceptor membranous disks, less twisted capillaries, improved cell morphology Biochemical: increased expression of P-occludin and decreased expression of ROCK1, P-MLC, VEGF	*F. lycii* demonstrated protective effects on blood-retinal barrier in diabetic rats by regulating the Rho/ROCK1 signaling pathway.
**B HUMAN CELL LINES**					
9	Song et al. (2011) [72]	Retinal ARPE-19 cell line incubated in high glucose	*F. lycii* extraction (methanol, ethanol, water, methanol/water)	Biochemical: Enhanced PPAR-γ luciferase activity along with PPAR-γ mRNA and protein expression (dose dependent effect), down-regulated the mRNA of pro-inflammatory mediators encoding MMP-9, fibronectin and protein expression of COX-2 and iNOS.	*F. lycii* extract and its associated taurine content enhanced PPAR-γ activation in retinal cells, providing a rationale for its use in prevention of DR. Methanol extract showed highest PPAR-γ activation when compared to other extracts.
10	Song et al. (2012) [73]	Retinal ARPE-19 cell line exposed to high glucose	*F. lycii* extract rich in taurine	Structural: Enhanced cell viability and decreased number of apoptotic cell Functional: attenuated high glucose induced apoptosis Biochemical: down-regulated caspase-3 protein expression and caspase-3 enzymatic activity.	*F. lycii* extract rich in taurine exhibited a cytoprotective effect against glucose exposure in a human RPE cell line in a dose dependent manner, probably due to reversal of caspase-dependent apoptotic cytotoxic pathway.
11	Pavan et al. (2014) [74]	Human RPE cells treated with 25mM glucose	Methanolic extract from *F. lycii*	Functional changes: Reversed decrease in trans epithelial electrical resistance, increased activity of cytosolic adenylyl cyclase Biochemical: increased intracellular cAMP levels	High glucose-induced barrier impairment of human RPE was ameliorated by treatment with *F. lycii* extracts through modulation of cAMP levels.

RPE: Retinal pigment epithelium; BiP: Binding immunoglobin protein; PERK: Extracellular signal-regulated kinase; ATF6: Activating Transcription Factor 6; AMPK: AMP-activated protein kinase; FOX03: Forkhead box 03 alpha factor; MnSOD: Manganese superoxide dismutase; ERG: Electroretinogram; NRF1: Nuclear respiratory factor; TFAM: Transcription factor A, mitochondria; HIF: Hypoxia induced factor: VEGF: vascular endothelial growth factor; HSP60: Heat shock protein 60; SOD: Superoxidase dismutase; MDA: Malondialdehyde; CRV: Central retinal vein; GFAP: Glial fibrillary acid protein; PEDF: pigment epithelium derived factor; ROCK: Rho kinase; P-MLC: Phospho myosin light chain; PPAR: Peroxisome proliferator activated receptor; MMP: Matrix metalloproteinase; COX-2: Cylcooxygenase 2; iNOS: inducible nitric oxide synthetase; cAMP: cyclic AMP.

**Table 4 nutrients-13-00246-t004:** List of studies that have evaluated *F. lycii* and retintis pigmentosa.

SNo	Authors (year)	Animal Model	*F. lycii* Administration	Impact on Variables	Conclusion
**A**	**STUDIES IN ANIMAL MODELS/CELL LINES**				
1	Wang et al. (2014) [102]	Mice (Wild-type rd10)	*F. lycii* (1 mg/kg, oral administration) post natal day 14, 25, 29 and 41	Structural: preserved outer retinal thickness and morphology of rods and cones, maintained ramified shape of microglial cells Functional: decreased latency and increased amplitude of b wave (photopic), larger scotopic a and b waves, improved visual behavior Biochemical: decreased levels of TNFα, IL-6β, CCL2, HIF-1α, activated caspase 3/7 and Bax, increased GSH/GSSG ratio	*F. lycii* exhibited long term neuroprotective effect on morphology and function of photoreceptors that was exerted through multiple pathways, such as antioxidation, antiinflammation and anti-apoptosis.
2	Zhu et al. (2016) [103]	Rats (Male Sprague-Dawley), N-Methyl-N-nitrosourea injection of 60 mg/kg	*F. lycii* (100, 200 and 400 mg/kg, gastric gavage) for 8 days or 14 days	Structural: Improved arrangement of outer nuclear and photoreceptor cell layer, maintained outer retinal thickness of central retina, decreased apoptotic cell ratio Biochemical: significant down-regulation of 9,3,7 of pro and cleaved caspases, increased nuclear levels of PARP and cleaved PARP protein	*F. lycii* attenuated methyl-N-nitrosourea induced photoreceptor cell apoptosis and protected retinal structure through regulation of PARP and caspase expression.
3	Liu et al. (2018) [101]	Mice (Wild-type rd1)	*F. lycii* (10 mg/kg, Intra-peritoneal injection) post natal day 4 to day 14, 20 or 24	Structural: Improved photoreceptor survival, restored morphology of rod and cone bipolar cells Functional: enhanced light evoked responses (rate, sensitivity and speed) of ganglion cells, increased b-wave amplitude, decreased abnormally high spontaneous spiking, protected ON pathway at early stages and OF pathway at later stages, improved visual behavior	*F. lycii* improved visual processing at multiple stages of information transmission via protecting morphology and function of photoreceptor and bipolar cells, function of ganglion cells, and perhaps, function of higher brain centers as suggested by enhanced visual behavior.
**B**	**STUDIES IN HUMAN SUBJECTS**				
4	Chan et al. (2019) [104]	Human subjects (*n* = 42) Double masked, placebo controlled	Oral administration of *F. lycii* in the form of granules (2 packs per day, each containing 5 g net wt) for 12 months	Structural: Preserved macular thickness Functional: Absence of deterioration of VA, both high contrast (90%) as well as low contrast (10%), no significant difference in visual field sensitivity or in any parameter of full field elctroretinogram	There was preservation of VA and macular structure following *F. lycii* supplementation over 12 month period, which is probably attributable to delay or minimization of cone degeneration due to neuroprotective effect of *F. lycii* on the retina.

TNF-α: Tumour necrosis factor alpha; IL-6β: Interleukin 6 beta; CCL-2: C-C motif chemokine ligand two; HIF-1α: Hypoxia-inducible factor one alpha; GSH/GSSG: Glutathione redox/antioxidant ratio; PARP: Poly (ADP-ribose) polymerase; VA: Visual acuity.

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
