# Peer review of "Fructus lycii*: A Natural Dietary Supplement for Amelioration of Retinal Diseases"

_nutrients, 2021, doi:10.3390/nu13010246_

Round 1

Reviewer 1 Report

Neelam and coauthors submitted a review article describing the role of Fructus lycii as an indigenous dietary supplement for amelioration of retinal diseases.  They did a very comprehensive review of the papers related the protective function of Fructus lycii in 3 retinal diseases: AMD, DR and retinitis pigmentosa.  This paper is well written and the readers will be interested in it.  

A minor comment

Retinitis pigmentosa is a group of heterogenous diseases due to genetic mutation. The pathogenesis of RP is  still not well understood. The authors only described the anti-oxidant function of Fructus lycii for PR.  In this erview article, the authors have to describe all the possible mechanisms of RP and point out the  oxidative damage might be the major mechanisn to cause retinal damage. Then they can explain the rational to use Fructus lycii in patients with RP.

Author Response

All authors have revised the manuscript titled “Fructus lycii: An indigenous dietary supplement for amelioration of retinal diseases”. In this point-by-point response we have addressed all the reviewers’ comments, where possible.  The reviewer’s comments are in automatic (black) and our responses are in automatic boldface. 

Reviewer 1

Neelam and coauthors submitted a review article describing the role of Fructus lycii as an indigenous dietary supplement for amelioration of retinal diseases.  They did a very comprehensive review of the papers related the protective function of Fructus lycii in 3 retinal diseases: AMD, DR and retinitis pigmentosa.  This paper is well written and the readers will be interested in it.  

A minor comment

Retinitis pigmentosa is a group of heterogenous diseases due to genetic mutation. The pathogenesis of RP is still not well understood. The authors only described the anti-oxidant function of Fructus lycii for PR.  In this review article, the authors have to describe all the possible mechanisms of RP and point out the oxidative damage might be the major mechanism to cause retinal damage. Then they can explain the rational to use Fructus lycii in patients with RP.

As requested by the reviewer, we have now included the information on possible mechanisms of RP and oxidative damage as a major mechanism to cause retinal damage (section 4.3, first paragraph, second and third sentences)

Reviewer 2 Report

In this review, Kumari Neelam et.al., explains the role of F lycii as an indigenous dietary supplement for amelioration of retinal diseases such as AMD, DR and RP. This review is an elaborate compilation of related literature, as this review summarized the existing literature on the sources of F lycii, bioactive components of F lycii and their bioavailability, background about AMD, DR, RP and therapeutic interventions of F lycii in pathogenesis of these retinal diseases. Although the topic of this review is quite interesting and of scientific relevance, the manuscript is quite confusing, with the concepts needing to be revised, using appropriate references in AMD, DR and RP sections. Importantly syntax and grammar must be reviewed, along with further use of abbreviations, as large parts of the manuscript are difficult to understand. A better structure should be defined to present “FRUCTUS LYCII AND RETINAL DISEASES AND DEGENERATION” reviewed by the authors. Also, rather than just including a concluding statement of the referenced articles in the review, author should furthur explain and discuss the statements. Finally, the tables could be more concise, and new figures could be added to sections such as bioactive components of F lycii and conclusion, with well-defined figure captions. A conclusion figure summarizing the molecular targets of F lycii in amelioration of retinal diseases will help the readers to comprehend the review well.

Author Response

All authors have revised the manuscript titled “Fructus lycii: An indigenous dietary supplement for amelioration of retinal diseases”. In this point-by-point response we have addressed all the reviewers’ comments, where possible.  The reviewer’s comments are in automatic (black) and our responses are in automatic boldface. 

Reviewer 2

In this review, Kumari Neelam et.al., explains the role of F lycii as an indigenous dietary supplement for amelioration of retinal diseases such as AMD, DR and RP. This review is an elaborate compilation of related literature, as this review summarized the existing literature on the sources of F lycii, bioactive components of F lycii and their bioavailability, background about AMD, DR, RP and therapeutic interventions of F lycii in pathogenesis of these retinal diseases. Although the topic of this review is quite interesting and of scientific relevance, the manuscript is quite confusing, with the concepts needing to be revised, using appropriate references in AMD, DR and RP sections.

Major revision has been made in terms of concepts for section on AMD, DR and RP (section 4, subsections 4.1, 4.2, 4.3, marked version of the article)

Importantly syntax and grammar must be reviewed, along with further use of abbreviations, as large parts of the manuscript are difficult to understand.

We have revised the entire manuscript with careful consideration of syntax, grammar and further use of abbreviations (marked version of the article). We have also included a ‘list of abbreviation’ in the form of table at the end of the manuscript.

A better structure should be defined to present “FRUCTUS LYCII AND RETINAL DISEASES AND DEGENERATION” reviewed by the authors.

Section 4 ‘Fructus lycii and retinal diseases and degeneration’ has been modified substantially in terms of structure for easy readability (section 4, subsections 4.1, 4.2, 4.3, marked version of the article).  

Also, rather than just including a concluding statement of the referenced articles in the review, author should further explain and discuss the statements.

We have now included detailed information on many subsections of AMD, DR and RP (section 4, subsections 4.1, 4.2, 4.3, marked version of the article)

Finally, the tables could be more concise, and new figures could be added to sections such as bioactive components of F lycii and conclusion, with well-defined figure captions. A conclusion figure summarizing the molecular targets of F lycii in amelioration of retinal diseases will help the readers to comprehend the review well.

All Tables have been made more concise by deleting unnecessary or duplicate information (tables unmarked copy).

New figure has been added to the section on bioactive components of F lycii (Section 2, figure 2 between second and third paragraph). In addition, a figure has also been added showing the molecular targets of F lycii in amelioration of retinal diseases (Section 7, figure 3 between first and second paragraph)

Round 2

Reviewer 2 Report

The authors have thoroughly addressed all the concerns raised in the initial manuscript and the efforts are greatly appreciated. The revised manuscript can deserve publication after making the minor changes mentioned below.

Minor changes:-

-Convert Figure 3 into a mind map diagram, as it looks like a table at the moment.

-Mention Figure 2 and Figure 3 in the manucript with a sentence.

-Line 49, change F-Lycii to F Lycii.

-Line 139, Change "increase bioavailability" to "increase in bioavailability"

Author Response

All authors have revised the manuscript and in this point-by-point response we have addressed all the reviewer’s comments.  The reviewer’s comments are in automatic (black) and our responses are in automatic boldface. 

Minor changes:

-Convert Figure 3 into a mind map diagram, as it looks like a table at the moment.

Figure 3 is now converted into mind map diagram

-Mention Figure 2 and Figure 3 in the manuscript with a sentence.

Figure 2 and 3 has been mentioned in the manuscript.

-Line 49, change F-Lycii to F Lycii.

Changed F-Lycii to F Lycii 

-Line 139, Change "increase bioavailability" to "increase in bioavailability"

Changed "increase bioavailability" to "increase in bioavailability"